# Numerical Simulation of the Post-Tensioned Beams Behaviour Under Impulse Forces Loading

**DOI:** 10.3390/ma18235432

**Published:** 2025-12-02

**Authors:** Anna Jancy, Adam Stolarski

**Affiliations:** Faculty of Civil Engineering and Geodesy, Military University of Technology, 2 gen. Sylwestra Kaliskiego Street, 00-908 Warsaw, Poland

**Keywords:** post-tensioned beams, finite element method analysis, concrete damage analysis, dynamic analysis, force impulse load

## Abstract

The paper presents the results of numerical simulation of the dynamic behaviour of the post-tensioned beams subjected to a constant force impulse load over time and a short-term force impulse load varying over time. Abaqus programme was used for numerical analysis, introducing necessary and detailed modifications to the modelling and calibration parameters. The numerical dynamics models were calibrated using results previously obtained from our own experimental and numerical static analysis. To estimate the dynamic strength of structural materials, the dynamic strength coefficient was applied in the concrete damage plasticity model, and the Johnson–Cook model was used to describe the evolution of the dynamic yield strength of steel elements. An explicit procedure was used to solve the dynamic equilibrium equations. The selection of the Rayleigh damping parameter and the methodology for determining the external load in a dynamic problem are discussed. The study presents new results on the influence of the type of force impulse loading and variable prestressing eccentricity in numerical simulations of post-tensioned beams. The results of the simulation show that the post-tensioned beams achieved a lower dynamic load capacity under a constant force impulse load of approximately 5% compared to the static load capacity achieved in the experimental static tests, regardless of the assumed prestressing eccentricity. A dynamic load capacity significantly exceeded the static load capacity under short-term time-varying force impulse loading. The beam with the larger prestressing eccentricity achieved a dynamic load capacity of 211% of the static load capacity, while the beam with the smaller prestressing eccentricity achieved a dynamic load capacity of 198% of the static load capacity.

## 1. Introduction

In typical structural design, most loads are considered static, especially those resulting from the self-weight of the structure, finishes, or live loads. However, dynamic loads are another type of load that significantly affects the design of structures. In many respects, dynamic loads differ from typical static loads. The response of a structure to dynamic loads is influenced by its inertia, damping and stiffness. Furthermore, there are some specific situations where dynamic actions must be considered. These include, for example, seismic and wind loads, undamped machine vibrations, or impact and blast loads. If the dynamic effects of these loads are not properly considered, it may lead to structural failure and a threat to human life.

To mitigate the risk of catastrophic structural failure, it is essential to understand how structures respond to dynamic actions. The dynamic behaviour of a structural system is influenced by both types of structural material and by the analytical approach used. Reinforced concrete represents a distinct composite construction material because it consists of two materials with completely different mechanical characteristics. To increase the load capacity of reinforced concrete elements, additional prestressing tendons are introduced. These prestressed tendons introduce additional compressive force into the elements. The resulting material, known as prestressed concrete, has improved performance under various loading conditions. Its behaviour under dynamic actions has been investigated by several researchers through experimental verification or numerical modelling.

Hop [1] analysed the influence of the prestressing ratio and the age of the elements on the natural frequency of post-tensioned beams with variable prestressing eccentricity. The study included beams made of both normal concrete and lightweight concrete, and their behaviour was verified by experimental studies. The author found that the age of post-tensioned concrete beams has a significant effect on the reduction in the logarithmic decrement of free vibration damping.

Noh et al. [2] investigated the effect of the prestressing force on the dynamic properties of post-tensioned concrete structures. The analysis included beams with different prestressing tendon arrangements subjected to impact tests. The study showed that natural frequency increases in the case of eccentrically post-tensioned beams, whereas no such relationship was found in the case of axially prestressed beams.

Kim et al. [3] conducted experimental tests on post-tensioned concrete beams in a bonded tendon system subjected to longitudinal cyclic impact loading. The aim of their research was to evaluate the influence of the prestressing force value on the natural frequency of beams, the elastic wave velocity, and the deformation modulus of the concrete. The studies confirmed that dynamic values of these parameters increase non-linearly depending on the prestressing level.

Cigada et al. [4] investigated the feasibility of detecting loss of prestressing force and failure of the prestressing strand using modal analysis. For this purpose, experimental tests were carried out on the dynamic behaviour of post-tensioned concrete beams with a system of straight, unbonded prestressing tendons. The study confirmed the difficulties of directly extrapolating the prestress conditions to single modal parameters, while it was more advantageous to consider all modal parameters together.

Kang et al. [5] investigated the response of seven post-tensioned beams subjected to repeated impact loading. The load was applied using a drop-weight device, which allowed for the application of repeatable impact loads. The study found that post-tensioned beams under repeated impact loading exhibited lower reaction forces during the second impact due to stiffness degradation and increased crushing. It has been concluded that flexural members maintain prestress more effectively than shear-type members, highlighting the importance of shear reinforcement and prestress maintenance in resisting repeated low-velocity impacts.

Sadraie et al. [6] analysed the dynamic performance of reinforced and post-tensioned concrete slabs with a constant prestressing eccentricity subjected to impact loading. Impact forces were generated by dropping weights from different heights. To compare the experimental results obtained, a numerical simulation was performed in the LS-DYNA programme. The authors modelled beam supports, columns, drop weights, and concrete slabs with 8-node solid elements. Steel reinforcement and prestressed strands were modelled with truss elements. The Winfrith concrete model with the shear failure surface proposed by Ottosen was used in the numerical simulation. Good agreement was obtained between the numerical simulation results and experimental results in terms of maximum acceleration, deflection, impact forces, and type of failure.

Tran et al. [7] investigated the impact performance of segmental concrete beams (PSCBs) made of geopolymer concrete (GPC) and prestressed using fibre-reinforced polymer FRP and steel tendons. Constant prestressing eccentricity was assumed in tests. The impact load was applied via a testing machine equipped with an impact head operating at an average impact velocity of 0.4 m/s, 1.5 m/s and 3.13 m/s. Based on experimental results, the authors proposed a simplified analytical model to predict the impact response of PSCBs. Based on the proposed analytical model, the load-deflection curve, global stiffness, impact force, and displacement under impact loads of simply supported segmental beams can be estimated.

Al Rawi et al. [8] investigated the behaviour of post-tensioned slabs subjected to impact loading induced by a freely dropped weight. For comparative analysis, reinforced concrete slabs with load-bearing capacities equivalent to those of post-tensioned specimens were prepared and tested under similar conditions. The authors have summarised the behaviour of the post-tensioned slabs under dynamic impact load due to the freely falling block, and differences in behaviour of reinforced concrete slabs by comparing displacements, impact force, cracks, and damage type.

Jiang and Chorzepa [9] provided an effective analysis procedure to analyse the failure of prestressed concrete beams with a constant prestressing eccentricity subjected to lateral impact loads using the LS-DYNA programme. Spatial finite elements were used to model the concrete, and the truss elements were used to model a steel reinforcement and prestressing tendon. The Winfrith concrete model in the LS-DYNA programme was used, and the Cowper–Symonds model was selected for the steel elements. The authors obtained good agreement between numerical results and experimental results.

Nghiem and Kang [10] have validated the available tools for non-linear finite element dynamic analysis (NLFEDA) in relation to the modelling of post-tensioned concrete beams with constant prestressing eccentricity under low-velocity impact. The authors have used the LS-DYNA software in which concrete elements were modelled as solid FE, and all steel reinforcement was modelled with beam elements. CSCM_CONCRETE model from the LS-DYNA library was used to reflect the concrete material behaviour. The model includes the effects of the strain rate to increase the strength of the concrete. PLASTIC KINEMATIC material was used for mild steel reinforcement, and the SPOTWELD model was used for unbonded tendons modelling. Both models included isotropic hardening plasticity; however, these did not include plastic hardening due to high strain rates. The authors confirmed the substantiated use of commercial software and material models with respect to the inertial effects of low-velocity impact loading of post-tensioned beams.

However, current research on reinforced concrete members subjected to short-term impact loading is more advanced, both in experimental tests and numerical modelling.

Mortar et al. [11] performed an impulsive load test to verify the behaviour of the Si-Al geopolymer concrete beam reinforced with steel bars. Based on the experimental tests, the numerical model was performed in Abaqus software. The concrete elements were modelled as C3D8R FE, and the Concrete Damage Plasticity (CDP) model was used. The CDP model was not modified to reflect the dynamic strength increase in concrete. For steel bars, the Johnson–Cook model was used. A good correlation between the experimental and numerical results was obtained.

Erdem and Gücüyen [12] have investigated the effect of impact loading on reinforced concrete beams with two different stirrup configurations. In addition, a carbon fibre reinforced polymer was used to strengthen the beams. The analysis was performed in Abaqus/Explicit. Concrete was modelled with C3D8R elements. Steel bars and CFRP strips were modelled with two-node beam elements (B31). The CDP model was used with Mander’s stress–strain model in compression. An increase in the dynamic strength of concrete was assessed using the dynamic increase factor (DIF) in compression and tension, which included strain rate effects. A similar approach was used for steel material. The dynamic strength was reflected by multiplying the yield strength and the DIF. The models were validated with experimental tests. The authors found that CFRP strengthening improved the impact behaviour of reinforced concrete beams.

Wu [13] has analysed the behaviour of concrete beams reinforced with engineered cementitious composite (ECC) and traditionally reinforced concrete beams under impact loading. For this purpose, numerical models were prepared in LS-DYNA software. The concrete material was modelled with solid elements, while the steel reinforcement and ECC were modelled with beam elements. Karagozian and Case Concrete Mode (K&C) with a damage function was used to reflect the concrete behaviour. The assumed model considers the strain rate effect on the dynamic strength increase. A similar model was used for ECC; however, an additional calibration was performed to reflect its behaviour in the implemented model. Steel elements were modelled with the Cowper–Symonds model, which also includes strain rate effects. As a result, it was numerically confirmed that ECC reinforcement has improved the protection performance of the beams subjected to impact load by 12% or 55.4%, depending on the thickness of the ECC layer.

Based on the literature review, there is a limited number of tests related to the dynamic behaviour of post-tensioned beams. Most of them focus on impact loading that involves freely dropped weight or hammer impact tests. This type of study is much more developed in relation to reinforced concrete beams [11,12,13]. Various modelling techniques, material model approaches, and software are used. Some of the researchers [12] provide their own modifications in CDP Abaqus models to simulate a dynamic strength increase in materials.

There are also different studies focused on modification procedures to the CDP model, incorporating the strain rate effect on the compressive or tensile strength of concrete.

Ayhan and Lale [14] proposed their modification procedure of the CDP model to include the strain rate effect. The modification was incorporated for compression hardening and tension softening using the dynamic increase factor based on strain rate developed by Gebbeken and Greulich [15]. The proposed model was verified in Abaqus in a Split Hopkinson bar test. The authors obtained a good relationship between numerical simulations and experimental results in relation to the dynamic increase factor and failure modes.

Zhu et al. [16] developed a modification of the CDP model by incorporating a rate correlation of the initial modulus of elasticity and the damage factor to simulate the impact resistance behaviour of concrete. The calibration of the proposed model was performed for the reinforced concrete slab under low and medium velocity impacts (from 1 to 150 s^−1^). The results of the analysis of the proposed material model were compared with experimental tests and other material models from the LS-DYNA software.

The basis of every dynamic analysis is an approach to the definition of the dynamic parameters of the materials. These depend on the stain rate. To correlate the stain rate with strength dependencies, experimental tests are required. Many researchers have investigated the influence of the strain rate on the properties of concrete and mild steel.

Wu et al. [17] have performed concrete impact compression tests using a Hopkinson bar. The researchers analysed the deformation characteristics, fracture behaviour, and energy release patterns for different strengths of concrete under high strain rates. The strain rates within a range of 55 s^−1^ to 150 s^−1^ were considered. The authors have summarised that the compressive strength increases significantly with increasing strain rate.

Cadoni et al. [18] have investigated the influence of different elevated temperatures (20 °C, 200 °C, 400 °C, 600 °C) and high strain-rates (10^−3^ s^−1^, 250 s^−1^, 400 s^−1^, 800 s^−1^) on the behaviour of the B500B reinforcing steel. The Split Hopkinson Tensile Bar has been used for that purpose. The parameters of the two constitutive laws, Johnson–Cook and Cowper–Symonds, have been provided based on the experimental tests performed.

Filiatrault and Holleran [19] have obtained the stress–strain relationships for reinforcing steel bars and concrete cylinders under seismic strain rates (80 × 10^−6^ s^−1^, 0.005 s^−1^, 0.02 s^−1^, and 0.1 s^−1^). The influence of the various temperatures was also considered for −40 °C, −20 °C, and 20 °C. The samples of reinforcing bars had a nominal yield strength of 400 MPa and a 6 mm diameter. Concrete was tested on cylinder samples with a 38 mm diameter and 76 mm height. The nominal compressive strength of concrete samples was 20 MPa.

Despite advanced studies related to experimental tests and numerical modelling of dynamically loaded reinforced concrete beams, there is a noticeable lack of research on short-duration impulse loads in relation to post-tensioned beams’ behaviour. Although some studies propose the approach to numerical modelling of dynamically loaded post-tensioned beams [6,9,10], none of them presents the influence of variable prestressing eccentricity on the dynamic behaviour of the post-tensioned beams in numerical simulations. All numerical models [6,9,10] have steel reinforcement bars and prestressing tendons modelled as 2-node elements, which causes a local multiplication of the stiffness of the materials. Additionally, many researchers have considered the dynamic strength increase phenomenon for concrete and steel materials by use of various theoretical approaches [6,9,10,11,12,13]. Some of them proposed their own modifications of the existing concrete models [14,16]. Materials testing to obtain the stress–strain relationship for concrete is commonly analysed by researchers. There is less research available from experimental tests for reinforcing rebars; however, there is a noticeable gap in the definition of the dynamic parameters of the prestressing steel. The procedure of the numerical model calibration to estimate the dynamic load capacity of the post-tensioned beams with detailed Rayleigh damping analysis was also not presented by any researchers. As a result, it creates a research gap in the approach of numerical modelling of the post-tensioned beams under impulse loadings.

This research introduces a novel approach that directly addresses these gaps.

The aim of this paper is to present the results of numerical simulation of the dynamic behaviour of the post-tensioned beams subjected to a constant force impulse load over time and a short-term time-varying force impulse load. The numerical models were calibrated using results obtained from the static analysis performed by Jancy et al. [20]. The geometry of post-tensioned beams is based on experimental studies. The paper presents an approach to dynamic material properties estimation. To obtain the dynamic strength of concrete, the dynamic strength coefficient was used based on the methods of Stolarski [21] and Stolarski et al. [22]. The Johnson–Cook model [23,24] for steel elements was chosen to describe the evolution of the dynamic yield stress. To justify the use of the delay yield in the dynamic flow of an elastic/viscoplastic material, Perzyna’s constitutive law of elastic-viscoplasticity [25] was used. For numerical analyses of post-tensioned concrete beams, the explicit Abaqus procedure was used to solve the dynamic equilibrium equations. The paper also presents the selection of the Rayleigh damping parameter and the methodology for external load determination in a dynamic problem. This approach provides an efficient means for model calibration, as it does not rely on commonly used mass scaling. The explicit method is a particularly effective computational approach compared to the Newton–Raphson method (Abaqus/Standard), in which the computational matrix must be inverted at each time step. Detailed simulation results of the post-tensioned beams’ behaviour under a constant force impulse load over time and short-term time-varying force impulse are presented.

## 2. Numerical Models of Post-Tensioned Concrete Beams

### 2.1. Geometry of Post-Tensioned Concrete Beams

Two 3.2 m long, post-tensioned beams with a variable tendon arrangement were analysed. The cross-sectional dimensions of both beams are 150 mm × 220 mm. The tendon profile axis is straight and horizontal for the first 300 mm of its length measured from the end of the beam, and then changes parabolically up to the maximum eccentricity at mid-span of the beam. The maximum eccentricity measured from the centreline of the beams to the tendons centre was 25 mm for beam No. 1 and 55 mm for beam No. 2. Both beams were made from C40/50 concrete grade. Prestressing steel with a characteristic tensile strength of $f_{pk}$ = 1860 MPa was used. A single 7-wire tendon with a diameter of 15.2 mm (total cross-section of 139 mm^2^) was used to transfer prestressing force. Two 10 mm diameter bars from B500SP steel grade, class C, were located in the upper and lower parts of the section. Additionally, 6 mm diameter bars as stirrups were added at 105 mm spacing within the central part of the beams and at 60 mm in the support zones. More details on beam geometry and material parameters can be found at Jancy et al. [20].

### 2.2. Numerical Model Description

The numerical models of the beams consist of separate spatial C3D8R finite elements: concrete, a prestressing tendon, two anchorages, stirrups and longitudinal reinforcing bars, Figure 1.

The mesh size was adapted to the geometry of the tendon in each beam. It varied depending on the curvature layout of the prestressing tendon. Typically, a 20 mm × 20 mm mesh grid was provided at the end of the beam, near the anchorage zones. It was then reduced to a 10 mm × 10 mm grid between the loading blocks. Some areas in beam No. 2 with a higher prestressing eccentricity required a further reduction in the mesh size to 4 mm × 4 mm. The adopted mesh size allowed for the analysis of crack propagation in both beams. The finite element mesh was created in Hypermesh version 2025 software and then exported to Abaqus/Explicit version 2025 for computational analysis.

#### 2.2.1. Contact Parameters

Depending on the beam components considered, different contact parameters were defined. In general, a surface-to-surface contact discretisation method is used. The surfaces of the finite elements between the anchorage and the tendon, the anchorage and the concrete, the reinforcing bars and the concrete were modelled as permanently connected with no possibility of slip. The interaction between the outer part of the tendon finite elements and the inner part of the concrete finite elements was modelled using the kinematic frictional contact method with finite slip. According to the manufacturer’s specification of the post-tensioned system [26], the friction coefficient in the tangential direction is μ=0.06.

#### 2.2.2. Calibration of Numerical Models of Beams in Static Tests

The numerical models used for dynamic simulation were calibrated for static tests [20]. For beam No. 1, very good agreement between experimental and numerical results was obtained. The experimental limit load capacity was $Q_{u1}^{exp}$ = 80.90 kN at a vertical displacement value of $u_{1}^{exp}$ = 36.6 mm. In turn, the numerical limit load capacity was $Q_{u1}^{num}$ = 80.64 kN at displacement $u_{1}^{num}$ = 36.5 mm. The numerical limit load capacity was obtained with an accuracy of 0.32% and 0.27% for vertical displacements with respect to the experimental result.

Good agreement of the limit load capacity and a smaller agreement of displacements were obtained for beam No. 2.

The experimental limit load capacity was $Q_{u2}^{exp}$ = 94.80 kN at a vertical displacement value of $u_{2}^{exp}$ = 53.1 mm. Numerical limit load capacity was $Q_{u1}^{num}$ = 96.50 kN at displacement $u_{2}^{num}$ = 36.4 mm. The numerical limit load capacity was obtained with an accuracy of 1.79%, but the displacement achieved a much lower accuracy of 31.5% compared to the experimental result.

A detailed comparative analysis of the obtained static experimental results and numerical results can be found in the work of Jancy et al. [20].

### 2.3. Materials Model

#### 2.3.1. Concrete

The concrete damage plasticity (CDP) model from the Abaqus library [27] was used for the numerical analysis. The basic explanation of the parameters and analysis of the CDP model is presented in the papers of Brenkus et al. [28], Jankowiak et al. [29], Sümer and Aktaş [30], Szczecina and Winnicki [31], Voyiadjis and Taqieddin [32] and Yu et al. [33]. In our work, the CDP model parameters were calibrated using static experimental tests [20], and their appropriately adapted values were assumed to simulate the dynamic behaviour of the analysed post-tensioned beams. The basis for the formulation of the dynamic law of stress evolution as a function of strain in concrete was an elastic–plastic material model with non-linear hardening, linear softening and constant residual stresses for compression and an elastic material model with linear softening and constant residual stresses for tension, Figure 2. These specific concrete model characteristics were calibrated using static tests, so similar, but appropriately scaled model assumptions were made in dynamic analysis. A detailed description of the stress–strain relationship and formulas can be found in Jancy et al. [20].

According to, among overs, Stolarski [21], Stolarski et al. [22], the compressive strength of concrete increases under the influence of dynamic loads. Based on these works, it was assumed that the value of the dynamic strength coefficient of concrete is in the range of $k_{d}$ = (1.0; 2.25] depending on the value of the strain rate.

A simplified method of determining the dynamic compressive strength of concrete was adopted using the dynamic strength coefficient $k_{d}$, which modifies the static compressive strength of concrete according to the formula:(1)$$f_{c,d}=k_{d}f_{c}$$

In this paper, two values of the dynamic strength coefficient of concrete were assumed in the analysis: $k_{d}$ = 1.2 and $k_{d}$ = 1.4, which, according to [21,22], correspond to strain rates of $\dot{\varepsilon }≅0.01\;s^{-1}$ and $\dot{\varepsilon }≅1\;s^{-1}$.

Figure 3 presents stress–strain relationships for the concrete strength assumed in the static analysis ($k_{d}$ = 1.0) and for increased dynamic strengths ($k_{d}$ = 1.2 and $k_{d}$ = 1.4).

The values of all required dynamic strength and deformation parameters, such as concrete tensile strength, effective elastic modulus and limit strains of individual plastic deformation phases, were determined based on the mean concrete compressive strength $f_{cm}$. With respect to strength parameters, the formulas from the EN 1992-1-1 standard [34] were used, and the Cichorski and Stolarski methods [35] were used to determine the values of limit strains.

The assumed concrete parameters for each dynamic strength coefficient are presented in Table 1. Detailed stress–strain notations and formulas can be found in Jancy et al. [20].

This modified relationship of the concrete stress–strain evolution was the CDP model in Abaqus/Explicit [27]. Therefore, it was necessary to determine different concrete damage parameters for compression and tension. Depending on the dynamic strength coefficient of concrete, different dependencies of damage parameters as a function of strain were defined.

The graphs in Figure 4 show the damage parameters as functions of strain for each dynamic strength coefficient. The compressive damage parameters were defined from the beginning of the non-linear plastic hardening phase. Detailed formulas for the damage parameter-strain can be found in Jancy et al. [20].

Within the CDP model, it is also required to define other basic parameters, such as the dilation angle ψ, the eccentricity ε of the plastic flow potential function, the ratio of initial compressive strength in an equal biaxial stress state to the initial uniaxial compressive strength $\sigma_{b0}/\sigma_{c0}$, the ratio of the second stress invariant on the tensile meridian $K_{c}$, and the viscosity parameter μ. Based on the calibration of the spatial FE models under static loads [20], similar basic parameters of the CDP material model were adopted. It was assumed that the damage mechanism would be the same as for static loads; therefore, the dilation angle ψ = 56.3 was considered for the dynamic analysis. All other parameter values of the CDP model are presented in Table 2.

#### 2.3.2. Steel Elements

In the numerical analysis of the dynamic behaviour of post-tensioned concrete beams, the Johnson–Cook material model was used to describe the dynamic response of steel elements [23,24]. This model uses the Huber–Mises–Hencky yield surface and the associated plastic flow law. The Johnson–Cook model describes the evolution of dynamic yield strength resulting from the combined effects of:(1)isotropic hardening dependent on the effective plastic strain,(2)dynamic hardening dependent on the strain rate,(3)temperature.

The dynamic yield strength is defined by the following equation:(2)$$\sigma_{yd}=\left[A+B\cdot {\left(\varepsilon_{eff}^{pl}\right)}^{n}\right]\cdot \left[1+C\cdot ln\left(\frac{{\dot{\varepsilon }}_{eff}^{pl}}{{\dot{\varepsilon }}_{0}}\right)\right]\cdot \left(1-\tau_{T}^{m}\right)$$
where

A—initial, static yield strength: ${A=f}_{y}$,B—plastic hardening modulus,$\varepsilon_{eff}^{pl}$—effective plastic strain,n—plastic hardening parameter,C—dynamic hardening parameter due to strain rate,${\dot{\varepsilon }}_{eff}^{pl}$—effective plastic strain rate,${\dot{\varepsilon }}_{0}$—reference strain for which the dynamic yield strength is equal to the static yield strength: $\sigma_{yd}({\dot{\varepsilon }}_{0})=A=f_{y}$,$\tau_{T}$—dimensionless thermal coefficient,m—thermal softening parameter.

Dimensionless thermal coefficient $\tau_{T}$ is described by the relationship:(3)$$0<\tau_{T}=\frac{T-T_{trans}}{T_{melt}-T_{trans}}<1$$
where

T—current temperature at which the tests were performed,$T_{trans}$—transition temperature at or below which the measured material parameters must be determined,$T_{melt}$—melting temperature at which the material will melt or no longer demonstrate shear capacity.

The limits on the value of the $\tau_{T}$ coefficient described in condition (3) means that if $T\leq T_{trans}$, then $\tau_{T}=0$, while for $T\geq T_{melt}$, then $\tau_{T}=1$, and therefore $\sigma_{yd}=0$.

Table 3 summarises the parameters adopted for the Johnson–Cook model depending on the material type of the individual steel elements in post-tensioned concrete beams.

The determination of the modulus B and the plastic strengthening parameter n was performed based on approximations of the static stress–strain relationships for prestressing tendons, reinforcing steel, and anchorage steel, see Jancy et al. [20].

However, determining the dynamic hardening parameter C requires detailed analyses based on the approximation of the experimental relationships of the dynamic yield strength as a function of the strain rate obtained in dynamic tensile tests. Due to the lack of such experimental results for steel grades used in the analysed post-tensioned concrete beams, the values of these parameters were determined approximately in this study. For this purpose, an approximation of the analytical results obtained based on the constitutive law of elastic-viscoplasticity of Perzyna [25] was used:(4)$$\sigma \left(t\right)=\sigma_{0}\left[1+{\left(\frac{{\dot{\varepsilon }}^{\upsilon pl}}{\gamma^{*}}\right)}^{\frac{1}{\delta }}\right]=\sigma_{0}\left\{1+{\left[\frac{\left(\dot{\varepsilon }\left(t\right)-{\dot{\varepsilon }}_{0}\right)}{\gamma^{*}}\right]}^{\frac{1}{\delta }}\right\},\;\gamma^{*}=\frac{2}{\sqrt{3}}\gamma$$
where

$\sigma_{0}$—static yield strength,${\dot{\varepsilon }}^{\upsilon pl}$—elastic-viscoplastic part of the strain rate,γ—viscosity coefficient of steel,δ—material coefficient.

Assuming a constant strain rate $\dot{\varepsilon }\left(t\right)=const.\;>{\dot{\varepsilon }}_{0}$ and assuming the steel viscosity coefficient of $\gamma =40.4\;s^{-1}$ and material coefficient δ=5, as that quite well satisfy the consistency of Equation (4) with experimental results for so-called mild steels, it can be shown that the dynamic strengths determined from Equation (2) for the dynamic hardening parameter C=0.05 are in good agreement with the results obtained from Equation (4). A similar approximation procedure was used by Bąk and Stolarski [36] to justify the delay yield effect in the description of the dynamic flow of an elastic/visco-perfectly plastic material.

Experimental tests conducted by Cadoni et al. [18] were performed at significantly higher stain rates (250 s^−1^, 400 s^−1^, 800 s^−1^) than those considered in this analysis. For reinforcing steel B500B, these tests reported a value of the parameter C = 0.02143 and the parameter n = 0.720.

We note that using the procedure presented in this study, the parameter C = 0.05 was determined for reinforcing steel B500SPC, which is a value of a similar order to the parameter C in [18].

Due to the stress–strain characteristics of very high-strength prestressing steel, a lower value of the dynamic hardening parameter C was adopted.

A sensitivity analysis of the dynamic behaviour of the post-tensioned beams under short-term, time-varying impulse loading to changes in parameters C and n was carried out.

The parameter C was analysed in the range [0.025; 0.075]. Negligible changes were observed in the dynamic response of the beams. Namely, the maximum displacement value in the first loading and unloading phases remained practically unchanged, and the oscillation characteristics were consistent.

Similarly, the parameter n was analysed in the range [0.024; 0.24]. At the maximum parameter value n = 0.24, the maximum displacement in the first loading phase increased by less than 2% compared to the displacement obtained for the minimum value of the analysed parameter n = 0.024. However, at higher values of n = 0.24, the minimum displacement value in the first unloading phase was increased by 73% compared to the displacement obtained for the *n* = 0.024, with a simultaneous increase in the vibration damping of the beams.

### 2.4. Methodology for Determining External Load in Dynamic Problems

A four-point bending scheme was adopted for post-tensioned concrete beams under dynamic loading. The assumed force impulse characteristics are presented in Figure 5.

Two types of dynamic load change over time were considered:

(1)constant force impulse over an infinite period (Figure 5a): (5)$$Q\left(t\right)=Q_{0}=const.$$(2)short-term force impulse that changes linearly over a finite period (Figure 5b):
(6)$$Q\left(t\right)=\left\{\begin{array}{ll} Q_{0}\left(1-\frac{t}{t_{f}}\right) & 0\leq t\leq t_{f}\\ 0 & t\geq t_{f} \end{array}\right\}$$
where$Q_{0}=\kappa Q_{u}$—initial value of the force impulse,$Q_{u}=\{Q_{u1},\;Q_{u2}\}$—static load capacity of the analysed beams [20],$\kappa =(0.1;\;\kappa_{d})$—dynamic load coefficient,$\kappa_{d}$—dynamic load capacity factor of beams: $Q_{d}=\kappa_{d}Q_{u}$,$t_{f}$—dynamic load duration.

A total force impulse is determined with respect to the static load capacity of the post-tensioned beams. Figure 6 presents the system of forces applied to the beams.

Dynamic loading is characterised by a sudden increase in load over time. Therefore, applying only a nodal force causes the strain rates of the concrete material to exceed the velocity of the dilatation wave. For this reason, the surface load distribution was applied over two areas of dimensions 30 mm × 150 mm, whose central axes in the beam width direction are located on the axes of load application on the upper surface of the beam (Figure 7).

### 2.5. Prestressing Force Value

Total prestressing force, including initial losses, was *P_m_*_0_(*x*) = 185.76 kN, which was converted to the initial stress in tendon *σ_p_* = 1336.4 MPa and applied in the beam model. The long-term losses were not considered in the prestressing force calculations, because the tendons were ultimately prestressed on the day of the experimental static tests [20].

### 2.6. Boundary Conditions

It is important to properly select the required direction of constraint on displacements and rotations at the support points to reflect the actual beam behaviour obtained in experimental tests. During the experimental tests [20], the beams were supported on roller supports at both ends. It is assumed that similar behaviour will be simulated in numerical models; therefore, displacements in the vertical (y-direction) and out-of-plane (z-direction) directions are blocked. Movement and rotations on the supports along the beam are released.

### 2.7. Parameters of the Numerical Calculation Procedure

#### 2.7.1. Dynamic Equilibrium Equation for Dynamic Problems Analysis

An explicit procedure was used to solve the equation of motion, assuming a lumped mass matrix with a diagonal structure, see Wriggers [37], Jancy et al. [20]. The general form of the equation of motion is described as follows:(7)$$M\ddot{w}(t)+C\dot{w}(t)+Kw\left(t\right)=P(t)$$
where M, C are the nodal mass and damping matrices, K is the stiffness matrix and P(t) is the vector of time-dependent applied load.

Damping matrix C can be described as a combination of mass and stiffness matrix:(8)C=αM+βK
where α and β are the mass and structural damping parameters.

Equation (7) can be transformed into the accelerations at the beginning of the current time:(9)$$\ddot{w}\left(t\right)=M^{-1}\left[P\left(t\right)-C\dot{w}\left(t\right)-Kw\left(t\right)\right]$$
where $\ddot{w}\left(t\right)$, $\dot{w}\left(t\right)$, and $w\left(t\right)$ are the acceleration, velocity, and displacement vectors.

The central difference scheme is used to explicitly integrate the equations of motion over time. Within this scheme, the velocities and displacements are determined at the end of the next time increment t+∆t knowing all the kinematic conditions from the previous increments:(10)$$\dot{w}\left(t+∆t\right)=\dot{w}\left(t-∆t\right)+2∆t\ddot{w}(t)$$(11)$$w\left(t+∆t\right)=w\left(t\right)+∆t\frac{\dot{w}\left(t+∆t\right)+\dot{w}\left(t\right)}{2}$$

The initialisation of the central difference scheme requires introducing the initial conditions at the time $t_{0}=0$:(12)$$\ddot{w}\left(t_{0}\right)={\ddot{w}}_{0},\;\dot{w}\left(t_{0}\right)={\dot{w}}_{0},\;\dot{w}\left(t_{0}-∆t\right)={\dot{w}}_{0}-∆t{\ddot{w}}_{0}$$
where the relation (12)_3_ follows on the first-order accurate Taylor series expansion for the velocity.

Since the explicit method of integrating the equations of motion is conditionally stable, it is necessary to introduce a time step constraint with respect to the critical time step:(13)$$∆t\leq {∆t}_{crit}=\frac{∆L_{min}^{e}}{c_{d,max}}$$
where $∆L_{min}^{e}$ is the smallest characteristic length of the element in the FE-discretisation, $c_{d,max}=\sqrt{\frac{E}{\rho }}$ is the fastest dilatation wave velocity in the elastic material, E is Young’s modulus, and ρ is the mass density.

#### 2.7.2. Selection of the Damping Parameter in Solving Dynamic Problems

The essence of damping determination in dynamic models is to stabilise the system, so that oscillations disappear after a certain time. Rayleigh mass damping was used, modifying the value of the α parameter. This parameter was chosen to obtain the aperiodic damped motion after reaching 5 to 10 times the maximum displacement amplitudes. The mass damping parameter was analysed in the range α = [5; 50].

## 3. Results of Numerical Research of Dynamically Loaded PT Beams

To estimate the dynamic load capacity of structural elements, it is necessary to use the two-stage approach. Firstly, by using the static load capacity of beams as the initial load values for both constant force impulse over time and a short-term impulsive force varying over time, the problem of the dynamic behaviour of the beams with zero damping parameter was solved. Then, the mass damping parameter must be obtained by analysing the behaviour of the beams under the influence of a constant force impulse over time. Once load capacity and damping parameter are known, further analysis can be performed under various short-term time-varying force impulse loads.

### 3.1. The Effect of a Constant Impulsive Force over Time

#### 3.1.1. Determining the Dynamic Load Capacity of Post-Tensioned Beams

The initial value of the force impulse constant over time was assumed to be equal to the static load capacity from experimental tests $Q_{01}=Q_{u1}$ = 80.90 kN ($\kappa_{1}$ = 1.0) for beam No. 1 and $Q_{02}=Q_{u2}$ = 94.80 kN ($\kappa_{2}$ = 1.0) for beam No. 2 [20]. For both beams, these loads were defined in the numerical model as surface loads on given areas with values equal to $q_{u,1}$ = 8.99 MPa and $q_{u,2}$ = 10.53 MPa at each force application location. The time characteristics of the force impulse load over time are shown in Figure 8.

In the first stage of the vibration analysis of both beams, the mass-damping parameter α = 0 and the dynamic concrete strength coefficient of $k_{d}$ = 1.2 were assumed. Figure 9 shows the displacement-time relationship obtained from the analysis. The results are presented for the node located in the mid-span of each beam, which achieves the highest value of vertical displacements.

In beam No. 1, constant oscillations were observed until time t = 120 ms, after which the vibrations stopped together with a non-linear increase in displacement. This feature indicates a loss of system stability. In beam No. 2, the oscillations also disappeared due to a loss of stability after reaching the first minimum displacement amplitude.

The disappearance of oscillations was also observed in the displacement velocity-time graphs (Figure 10a). Upon reaching the first maximum amplitude, both beams had similar vibration velocities. In later phases, the displacement velocity decreased for beam No. 1 at time t = 120 ms and for beam No. 2 at time t = 50 ms.

Clear differences in acceleration change over time were observed in the diagrams for both beams. Beam No. 1 had very high acceleration values, whereas beam No. 2 experienced low acceleration values from the beginning of loading (Figure 10b). The acceleration values in Figure 10b are expressed in units of standard gravity g, where 1 g equals the value of the gravitational acceleration on Earth g = 9806.65 mm/s^2^.

Due to the loss of stability of both beams under dynamic loading corresponding to the static load capacity, an analysis was performed to find a load for which the solution does not demonstrate a loss of stability of the system. The values of the dynamic load factor were analysed in the range κ = [0.76; 1.0], considering the dynamic strength coefficient of concrete $k_{d}$ = 1.2. Figure 11 presents the results of the analysis of the displacement-time relationship for various values of the constant force impulse load over time for both beams.

The highest dynamic load values at which regular oscillations were achieved were $Q_{01}$ = 77 kN ($\kappa_{1}$ = 0.95) for beam no. 1 and $Q_{02}$ = 87 kN ($\kappa_{2}$ = 0.94) for beam no. 2. The maximum displacement value of the first amplitude is equal to $u_{1,max}$ = 41.31 mm and was achieved at the time $t_{a1}$ = 24 ms for beam no. 1. For beam no. 2, these values were $u_{2,max}$ = 37.15 mm at time $t_{a2}$ = 22.5 ms.

The load values determined in this way should be considered as the dynamic load capacity of the beams ${Q_{d1}=Q}_{01}$ = 77 kN and ${Q_{d2}=Q}_{02}$ = 87 kN. Regardless of prestressing eccentricity, the beams achieved a similar dynamic load factor κ = 0.95.

#### 3.1.2. Selection of the Mass Damping Parameter

Then, the numerical model of beam No. 1 was analysed for the values of the mass-damping parameter α = 5, α = 25 and α = 50 under a constant force impulse loading at a time corresponding to its dynamic capacity. A dynamic concrete strength coefficient $k_{d}$ = 1.2 was assumed for the analyses. The displacement-time diagrams are presented in Figure 12.

For the value of the assumed mass damping parameter α = 5, model oscillations were practically not reduced. For a mass damping parameter of α = 25, the value of the first maximum amplitude decreased by approximately 15% compared to the value obtained in the analyses without the specified mass damping parameter. However, the time to reach this amplitude remained unchanged. As a result, for the damping parameter α = 50, significant damping of the vibrations of beam No. 1 was achieved already after the third maximum displacement amplitude.

The numerical solution for the mass damping parameter α = 25 was found to be consistent with the assumed condition of vibration disappearance after reaching 5 to 10 maximum displacement amplitudes, as explained in Section 2.7.2 of this paper.

For beam No. 2, the use of the mass damping parameter *α* = 25 also resulted in vibration damping consistent with the assumption adopted for beam No. 1. Figure 13 presents the displacement to time diagram for beam No. 2 with and without the adopted damping parameter. The maximum value of the first amplitude was $u_{2,max}$ = 31.64 mm, which was achieved at the time $t_{a2}$ = 22.5 ms (Figure 13). The difference in the amplitude value compared to the model without damping was 15%, while the time to reach the first maximum amplitude remained similar.

#### 3.1.3. Influence Analysis of the Dynamic Strength Coefficient of Concrete on the Dynamic Response of Post-Tensioned Beams for Constant Force Impulse over Time

To analyse the effect of dynamic concrete compressive strength on the oscillation behaviour of the beams, dynamic strength coefficients $k_{d}$ = 1.2 and $k_{d}$ = 1.4 were used. The analyses were performed for a selected mass damping parameter of α = 25, for which the oscillations ceased after reaching the fifth maximum displacement amplitude (Figure 14). The assumed loading value was equal to the dynamic load capacity of the beams.

For beam No. 1, the time to reach the maximum displacement amplitudes did not change. The first maximum displacement amplitude was reached at $t_{a1}^{1.4fc}$ = 24 ms for $k_{d}$ = 1.4, similar to the case for $k_{d}$ = 1.2. The value of the first displacement amplitude increased to $u_{1,max}^{1.4fc}$ = 38.65 mm compared to the displacement of the beam with concrete strength increased by only 20% ($k_{d}$ = 1.2), for which the value of the first maximum amplitude was $u_{1,max}=u_{1,max}^{1.2fc}$ = 35.19 mm. The displacement difference was $∆u_{1,max}≅$ 3.46 mm. The dynamic capacity increased by 3% compared to the static load capacity ($Q_{01}=Q_{u1}$ = 80.90 kN) and was equal to $Q_{01}^{1.4fc}$ = 83 kN (Figure 14).

The dynamic capacity of beam No. 2 was $Q_{02}^{1.4fc}$ = 97 kN (Figure 14) for a dynamic strength coefficient of concrete $k_{d}$ = 1.4. Dynamic capacity has increased by 6% compared to its static capacity $Q_{02}=Q_{u2}$ = 94.80 kN. The first maximum displacement amplitude was reached at the same time $t_{a2}$ = 22.5 ms for both values of the dynamic strength coefficient of concrete $k_{d}$ = 1.2 and $k_{d}$ = 1.4. However, the displacement values were different. The first maximum amplitude with concrete strength increased by 40% was $u_{2,max}^{1.4fc}$ = 37.88 mm, while for concrete strength increased by 20% it was $u_{2,max}^{1.2fc}$ = 31.64 mm. The displacement difference was $∆u_{2,max}$ ≅ 6.24 mm.

Increasing the dynamic strength of concrete by using a dynamic strength coefficient does not significantly reduce the damped dynamic displacements of post-tensioned concrete beams loaded with a constant force impulse over time. However, it has a greater effect on increasing the dynamic load-bearing capacity of the beams.

### 3.2. The Effect of a Short-Term Time-Varying Force Impulse

#### 3.2.1. Determination of the Dynamic Load-Capacity of Post-Tensioned Beams

Based on the results obtained from the analysis of the behaviour of beams under the influence of a constant force impulse over time, the initial values of the parameters determining the time-varying force impulse load were estimated.

The maximum time-varying force impulse load was assumed to be the same for both beams: $Q_{d}$ = 160 kN. This value allows for the determination of the dynamic load factors for each beam as described in Equation (6): $\kappa_{1}=Q_{0}/Q_{u1}$ and $\kappa_{2}=Q_{0}/Q_{u2}$. The static capacity of post-tensioned beams from the experimental test was: $Q_{u1}$ = 80.90 kN and $Q_{u2}$ = 94.80 kN, the corresponding values of the dynamic load factors are: $\kappa_{1}$ = 1.98 and $\kappa_{2}$ = 1.69.

The duration of the short-time impulse load that varies linearly in time was assumed to be $t_{1}$ = 24 ms for both beams. This value of $t_{1}$ approximately corresponds to the time of reaching the first maximum displacement amplitude in both beams under constant force impulse load over time. Figure 15 shows the load characteristics of the short force impulse load varying in time.

The analyses were carried out assuming the mass damping parameter α = 25 and the dynamic strength coefficient of concrete $k_{d}$ = 1.2.

Figure 16 shows the displacement-time relationship for both beams under an assumed short-term time-varying force impulse load. The approximate load application time is also marked in grey in Figure 16.

In the first loading phase at time t = [0; 24] ms, deflections increased in the direction of the load. For beam No. 1, the maximum value of the first displacement amplitude occurred at time t = 21 ms and was equal to $u_{1d}$ = 54.13 mm. For beam No. 2, the maximum displacement value occurred at time t = 16.5 ms, while the maximum displacement value was $u_{2d}$ = 43.29 mm.

An unloading phase occurred after reaching the maximum value of the first displacement amplitude, then the system returned to the equilibrium position with permanent plastic deformations in the concrete. For beam No. 1, the first minimum displacement amplitude of $u_{1d}$ = 10.54 mm was reached at time t = 54 ms. After that, the beam oscillations reached very small values of the amplitude differences, velocities (Figure 17a), and accelerations (Figure 17b). In the case of beam No. 2, the system returned to the equilibrium position at time t = 45 ms, reaching a negative value of the first minimum displacement amplitude of $u_{2d}$ = −7.54 mm. Subsequent system vibrations also occurred with negative displacement amplitudes around the negative value of the permanent displacement.

As a result of the beam being unloaded, the concrete in the upper layers of the cross-section was destroyed, with a simultaneous predominance effect of a greater eccentricity of the prestressing force.

The change in displacement velocity reflects the vibration characteristics of both beams for critical phases in the process of reaching the first maximum displacement amplitude under loading and the first minimum displacement amplitude under the unloading phase (Figure 17a). The displacement velocity values for both beams are similar. After stabilisation of the system, displacement velocities oscillate around υ = 0 m/s with small velocity deviations Δυ ≅ ±0.5 m/s. Differences in progressive degradation of beams’ stiffness are also indicated by the characteristics of the acceleration-time diagrams (Figure 17b). Beam No. 1 achieved greater positive and negative acceleration (deceleration) over time than beam No. 2.

Differences in damage characteristics are visible in the stress distributions and the damage parameter in both compression and tension. When the first maximum displacement amplitude is reached in beam No. 1, a residual stress area is visible in the compression zone (Figure 18a). However, the compressive strength of concrete was not reached in the compression zone of beam No. 2 (Figure 18b).

During the unloading phase in both beams, the compression zone changed its position to the lower layers of the cross-section, and at the same time, residual compressive stresses appeared in this zone. This indicates the location of the damage in the lower layers of the beams (Figure 19).

Figure 20 shows the distribution of the damage parameter in compressed concrete for both beams. The distribution of this parameter value indicates that for beam No. 1, the capacity of the upper compression zone was lost during the first loading phase, while for beam No. 2, the damage parameter did not reach the maximum value.

During the first unloading phase, the extent of concrete compression failure was similar for both beams. In beam No. 1, both the upper and lower sections of the cross-section failed in compression after the unloading phase. However, the failure is localised only in the lower section of beam No. 2 (Figure 21).

The stress distribution in the reinforcing bars is similar in both beams in the first loading phase, but in the unloading phase (Figure 22), there are differences in the location of the maximum stresses.

In the first phase of unloading, the maximum stress value in the reinforcement bars of beam No. 1 was in the area between the support point and the point of force application. However, in beam No. 2, the maximum stress value occurred in the upper reinforcing bars in the area between the force application points. In both phases, the reinforcing bars locally reached values in the viscoplastic strain range.

The design assumption for both beams, based on the failure of the concrete compression zone, also influenced the damage pattern under dynamic loading. In the case of the prestressing cable, the strains also locally exceeded the yield limit (Figure 23) in both beams.

Due to the unbonded prestressing system, the stresses in the prestressing cable only increase due to the deformation of the beam.

#### 3.2.2. Damage Mechanism Analysis and Load-Capacity Analysis

The dynamic load capacity of the post-tensioned beams can be estimated based on the damage distribution in the first loading phase when the first maximum displacement amplitude is reached. If the compression zone is damaged during this phase, the load capacity will be reached. The dynamic load capacity defined in this way is achieved in beam No. 1 (Figure 21a), while in beam No. 2 (Figure 21b), this criterion was not met.

Taking this fact into account, the compression damage distribution of beam No. 2 was analysed for various values of the initial load with a short-term time-varying force impulse (Figure 24).

At a load value of $Q_{0}$ = 200 kN ($\kappa_{2}$ = 2.11), the damage distribution of beam No. 2 meets the assumed dynamic load-capacity criteria of a post-tensioned beam. The maximum displacement during the first loading phase was $u_{2d}$= 67.0 mm and occurred at t = 19.5 ms. During the first unloading phase, the first minimum displacement amplitude was negative and equal to $u_{2d}$ = −21.45 mm and was reached at t = 54 ms.

During the first loading phase, the lower layers of the cross-section were damaged. However, at the first unloading phase, concrete damage occurred in the upper layers of the cross-section of beam No. 2 (Figure 25).

#### 3.2.3. Influence Analysis of the Dynamic Strength Coefficient of Concrete on the Dynamic Response of Post-Tensioned Beams for Variable Force Impulse over Time

Similarly to the comparative analysis of the constant force impulse over time, two values of the dynamic strength coefficient of concrete were assumed $k_{d}$ = 1.2 and $k_{d}$ = 1.4. Both beams were loaded with the determined dynamic load-capacity values for the short-term time-varying force impulse, which were $Q_{0}$ = 160 kN ($\kappa_{1}$ = 1.98) for beam No. 1 and $Q_{0}$ = 200 kN ($\kappa_{2}$ = 2.11) for beam No. 2. The impulse load was also applied at time $t_{1}$ = 24 ms. The mass damping parameter α = 25 was assumed in the analyses.

In the case of the dynamic strength coefficient of concrete considered, increasing the strength value by 40% ($k_{d}$ = 1.4) significantly changed the dynamic response characteristics of both beams (Figure 26).

In beam No. 1, the value of the first maximum displacement amplitude reached $u_{1d,max}^{1.4fc}$ = 51.51 mm at time $t_{a1}^{1.4fc}$ = 18 ms, which is approximately 5% less than the first displacement amplitude in this beam with the dynamic strength coefficient of concrete of $k_{d}$ = 1.2. The time of reaching the first maximum amplitude occurred $\Delta t_{a1}$ = 3 ms earlier than in the beam with a lower dynamic strength.

The values of the difference in displacement amplitudes that occur after unloading the beam also decreased. In the case of the dynamic strength coefficient of concrete $k_{d}$ = 1.2, oscillations occurred in the range of displacement amplitude difference ${∆u}_{1d}^{1.2fc}≅$ 15 mm, while for higher dynamic concrete strength with $k_{d}$ = 1.4, oscillations occurred at a displacement amplitude difference of ${∆u}_{1d}^{1.4fc}≅$ 5 mm.

In beam No. 2, similar changes in displacement amplitudes occurred as a result of the applied different values of dynamic concrete strength, as in beam No. 1. In beam No. 2, the value of the first maximum displacement amplitude $u_{2d,max}^{1.4fc}$ = 62.64 mm, decreased by approximately 6.5% compared to the amplitude value $u_{2d,max}^{1.2fc}$ = 67 mm in this beam with a dynamic strength coefficient of concrete $k_{d}$ = 1.2. The time when the maximum displacement amplitude in beam No. 2 occurred was later for the higher concrete strength $k_{d}$ = 1.4: $t_{a2}^{1.4fc}$= 21 ms, than in the case of using a lower dynamic concrete strength coefficient $k_{d}$ = 1.2: $t_{a2}^{1.2fc}$= 19.5 ms.

However, the most important phenomenon that occurred with the increased strength of concrete in beam No. 2 is the lack of negative oscillations of displacement amplitude values in the unloading phase.

In compression damage characteristics observed for the higher dynamic concrete strength coefficient ($k_{d}$ = 1.4), there was no stiffness degradation in the finite elements located between the centre of gravity of the cross-section and the prestressing tendon. In the upper layers of the beam and in the central zone, the extent of the damage action also decreased.

## 4. Conclusions

This paper presents a numerical simulation of the dynamic behaviour of post-tensioned beams based on calibrated models to static experimental research [20].

Based on the results obtained, the following conclusions can be drawn, relevant for analysing dynamic behaviour and modelling of dynamically loaded post-tensioned beams using the FEM.

The most important technical findings resulting from the research are presented below.

The explicit method for solving the dynamic equilibrium equation using Abaqus/Explicit is an efficient numerical method for analysing the dynamic problem for post-tensioned beams with a non-linear prestressing tendon.The use of a Rayleigh mass damping parameter of α = 25 for post-tensioned beams with different prestressing eccentricity led to a damped motion that decayed after reaching the maximum displacement amplitude 5–10 times.Modelling the dynamic behaviour of concrete based on the assumption of a constant dynamic strength coefficient of concrete in compression.Numerical models of dynamically loaded post-tensioned concrete beams showed some specific differences in the influence of the prestressing eccentricity on the value of displacement change over time, the permanent displacement value and the dynamic load capacity. These values depend on the specific type of force impulse action.In the numerical simulation of the effects of a constant force impulse over time, the post-tensioned beams obtained a lower load capacity by approximately 5% compared to the load capacity achieved in the experimental static tests [20], regardless of the assumed prestressing eccentricity. A constant amplitude difference was achieved in both beams, which indicates the stability of the method. However, the values of permanent displacements were smaller in the beam with the higher prestressing eccentricity for a load value of 95% of the static load capacity.The value of the dynamic strength coefficient of concrete significantly influences the dynamic response characteristics of post-tensioned beams under a constant force impulse over time. The dynamic load capacity increases relative to the static load capacity for higher values of the dynamic strength coefficient of concrete. The beam with a larger eccentricity achieved a 6% increase in dynamic load capacity, while the beam with a smaller eccentricity achieved a 3% increase in dynamic load capacity.Numerical models of post-tensioned beams showed a dynamic load capacity significantly exceeding the static load capacity under a short-term time-varying force impulse load. The beam with the larger prestressing eccentricity achieved a dynamic load capacity at an initial impulse load value of 211% of the static load capacity, while the beam with the smaller prestressing eccentricity achieved a dynamic load capacity value of 198% of the static load capacity. During the loading phase, both beams had similar vertical displacement characteristics. However, the beam with the larger prestressing eccentricity reached the first maximum displacement amplitude earlier than the beam with the smaller prestressing eccentricity at the same initial short-term impulse load. In the unloading phase, as the system returned to the equilibrium position and permanent plastic deformations in the concrete were achieved, the beam with the greater prestressing eccentricity reached a negative value of the first minimum displacement amplitude. However, in the beam with the smaller prestressing eccentricity, the values of the first minimum displacement amplitude were positive.The value of the dynamic strength coefficient of concrete significantly influences the dynamic response characteristics of prestressed post-tensioned concrete beams under the action of a short-term time-varying force impulse load. When a higher value of the dynamic strength coefficient of concrete was applied to the beam with a larger prestressing eccentricity, no negative oscillations of displacement amplitudes occurred during the unloading phase. However, increasing the value of the dynamic strength coefficient of concrete in the beam with a smaller prestressing eccentricity resulted in only a reduction in permanent displacements by approximately 67%.

In turn, practical implications for the design and evaluation of structures can be presented as follows:The presented method for determining the dynamic load capacity of post-tensioned beams is applicable to other structural elements. On the basis of static load capacity, the initial dynamic loading conditions can be assumed for numerical simulations.A constant amplitude difference achieved under constant load over time and without damping indicates the stability of the analysis method.The dynamic load capacity increases relative to the static load capacity for higher values of the dynamic strength coefficient of concrete.Depending on the prestressing eccentricity, the post-tensioned beams indicated some specific differences in behaviour under short-term, time-varying impulse force. It was related to the predominant effect of the greater prestressing eccentricity during the unloading phase, in conjunction with changes in the concrete compression damage zones.The conducted numerical simulations show that post-tensioned beams with variable prestressing eccentricity effectively contribute to dynamic loading.

The presented results constitute the basis for predicting the parameters of experimental tests of post-tensioned beams subjected to force impulse loading, because:Appropriate parameters of the recording equipment used in experimental tests are essential for accurately capturing very high accelerations and displacements. In the case of short-term, time-varying impulse loading, these phenomena occur within extremely short time intervals.Preparation of the experimental stand should enable adjustments for large vertical displacement values, particularly observed in post-tensioned beams with larger prestressing eccentricity.The design of the experimental stand should include sufficiently stiff supports to prevent the occurrence of vertical displacements, as these influence the measured accelerations and displacement values.The design of the experimental stand must accommodate the vertical upward movement of post-tensioned beams at midspan, caused by inertial effects following load application.Accurate recording of the magnitudes of force impulse at all points of application is essential in four-point bending tests.Strain rate values affect the strength of the materials used in the test beams, including concrete, reinforcing steel, and prestressing steel. For each material component, these values must be determined through experimental testing.

## Figures and Tables

**Figure 1 materials-18-05432-f001:**
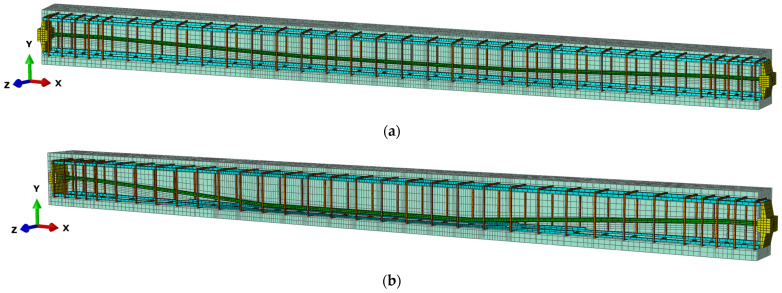
FE spatial computational model: (**a**) beam No. 1; (**b**) beam No. 2.

**Figure 2 materials-18-05432-f002:**
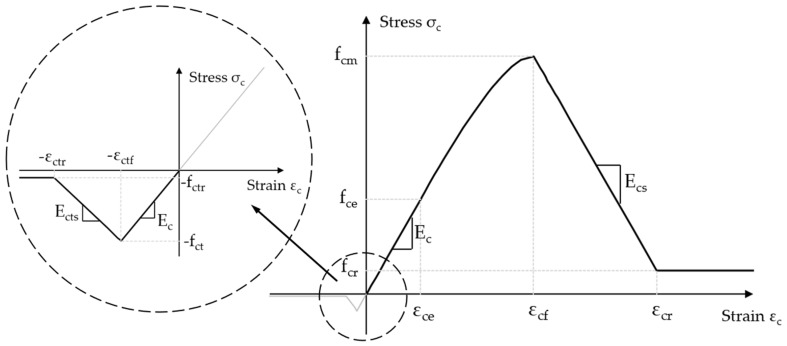
Description of compression and tension behaviour of concrete in the CDP model.

**Figure 3 materials-18-05432-f003:**
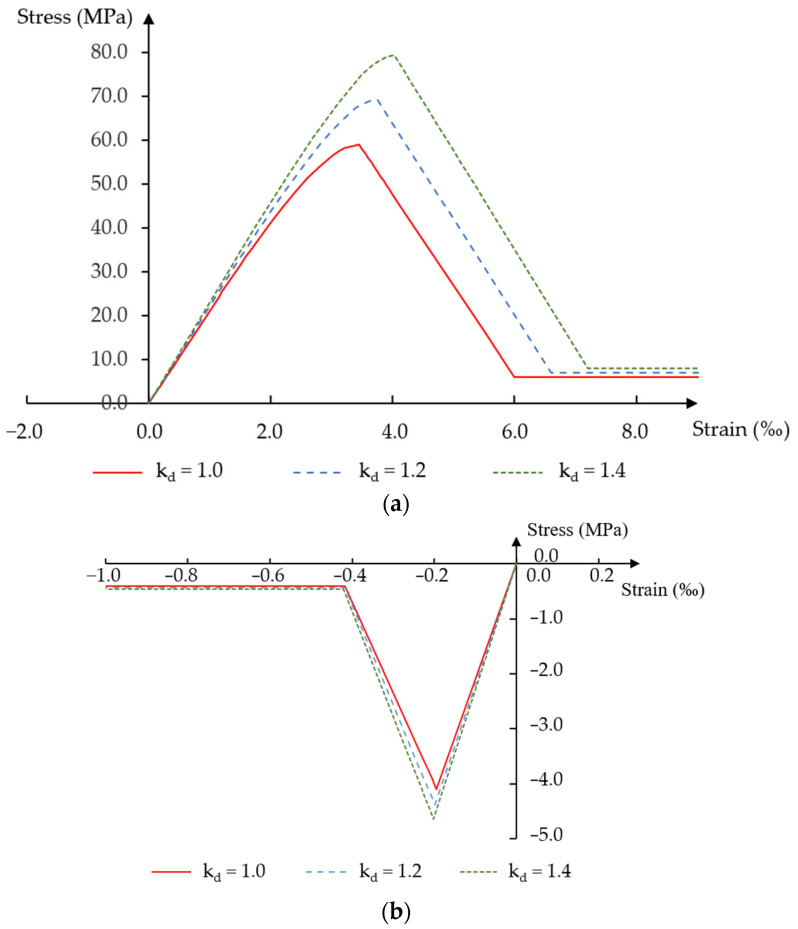
Stress–strain relationship of concrete for different values of the dynamic strength coefficient: (**a**) compression; (**b**) tension.

**Figure 4 materials-18-05432-f004:**
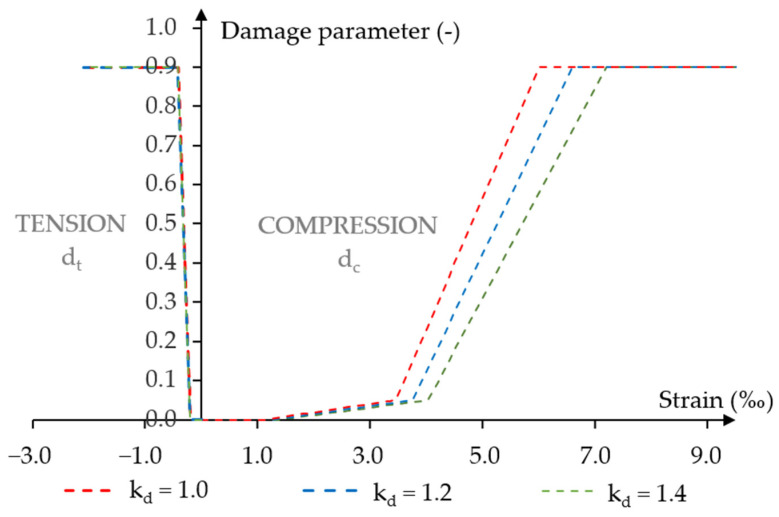
Concrete damage parameter-strain relationships for the assumed values of the dynamic concrete strength coefficient.

**Figure 5 materials-18-05432-f005:**
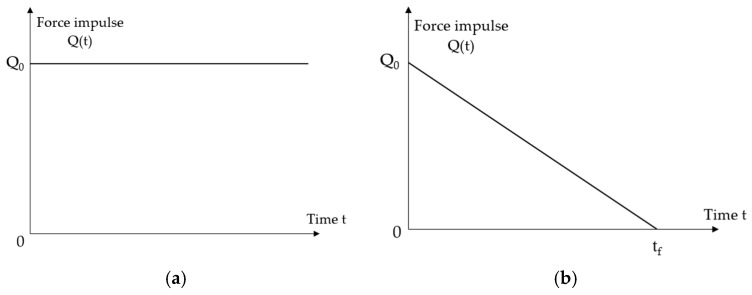
Force impulse loading characteristics: (**a**) constant over time; (**b**) variable over time.

**Figure 6 materials-18-05432-f006:**
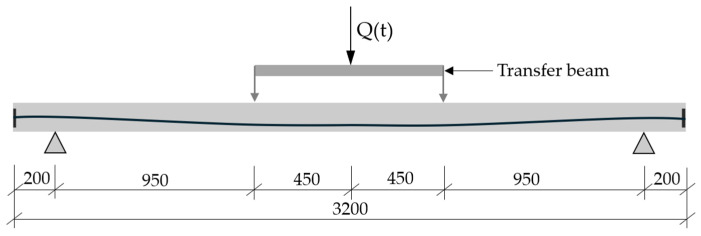
The system of forces loading the beams (dimensions in mm).

**Figure 7 materials-18-05432-f007:**
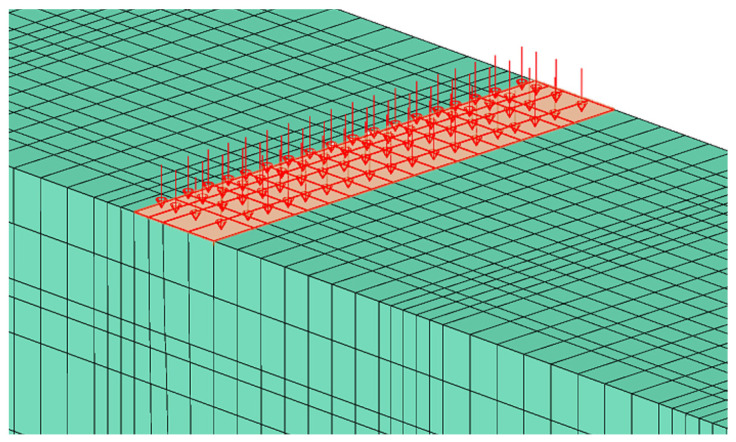
Surface loading with dynamic force.

**Figure 8 materials-18-05432-f008:**
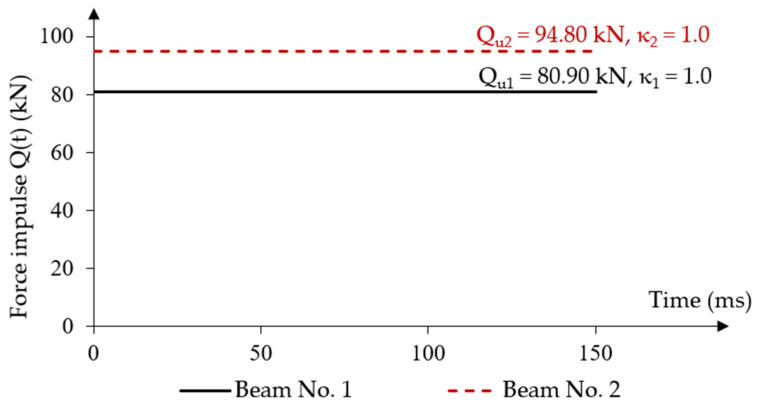
Characteristics of a constant force impulse over time.

**Figure 9 materials-18-05432-f009:**
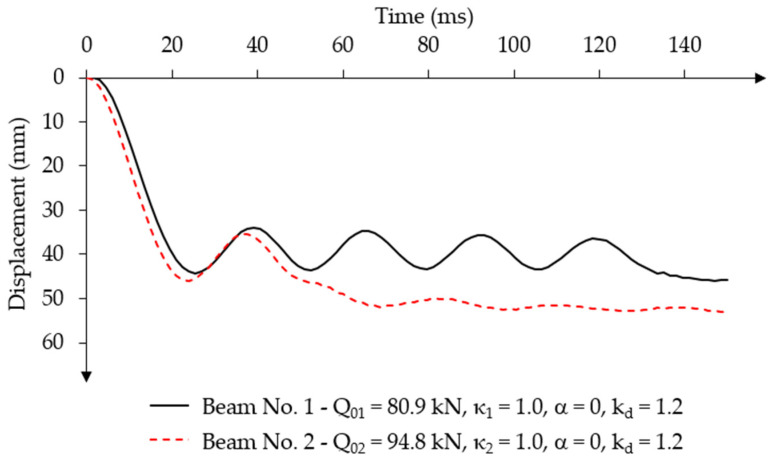
Displacement-time relationship for a constant force impulse load, constant over time, without damping.

**Figure 10 materials-18-05432-f010:**
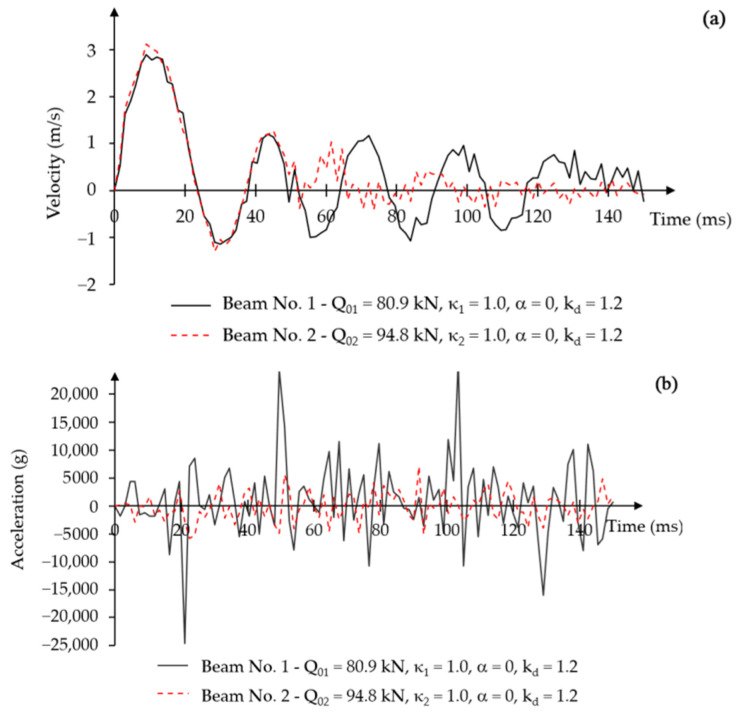
Force impulse load constant over time, without damping: (**a**) displacement velocity–time relationship; (**b**) acceleration–time relationship.

**Figure 11 materials-18-05432-f011:**
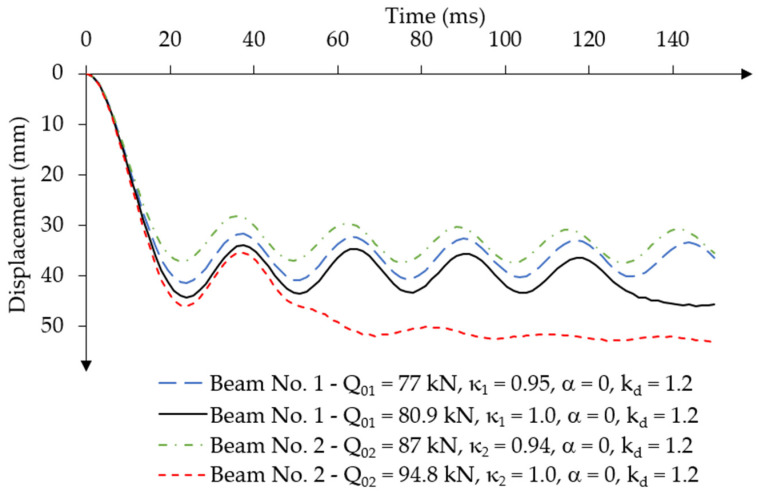
Results of the displacement-time relationship for different values of the constant force impulse load over time for Beam No. 1 and Beam No. 2, without damping.

**Figure 12 materials-18-05432-f012:**
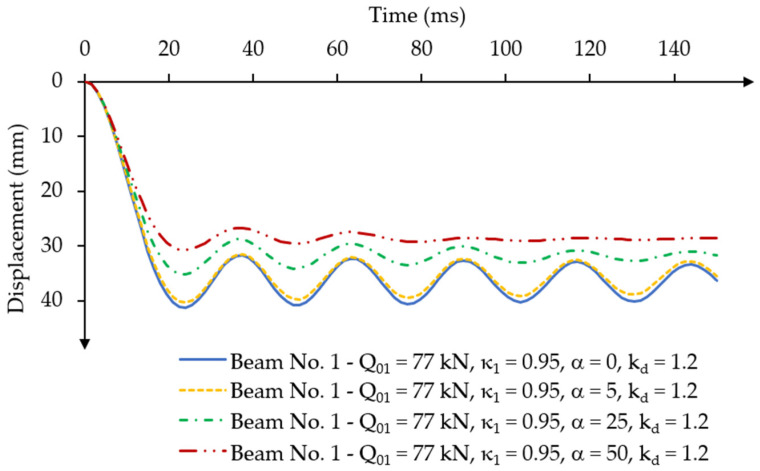
Results of the displacement-time relationship under constant force impulse loading over time for different values of the damping parameter for beam No. 1.

**Figure 13 materials-18-05432-f013:**
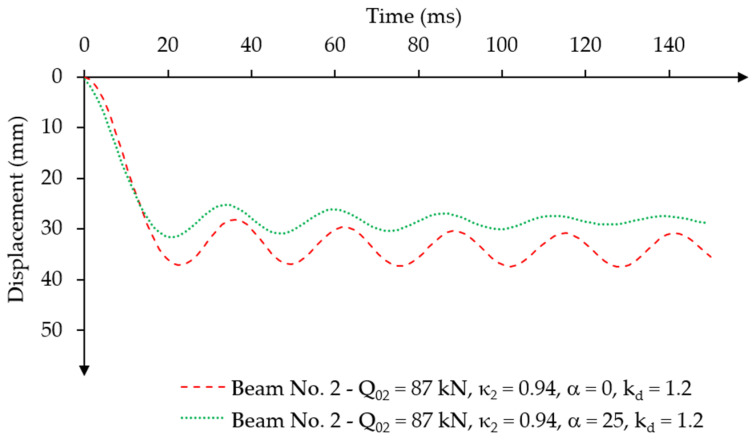
Results of the displacement-time relationship under constant force impulse loading over time for different values of the damping parameter for beam No. 2.

**Figure 14 materials-18-05432-f014:**
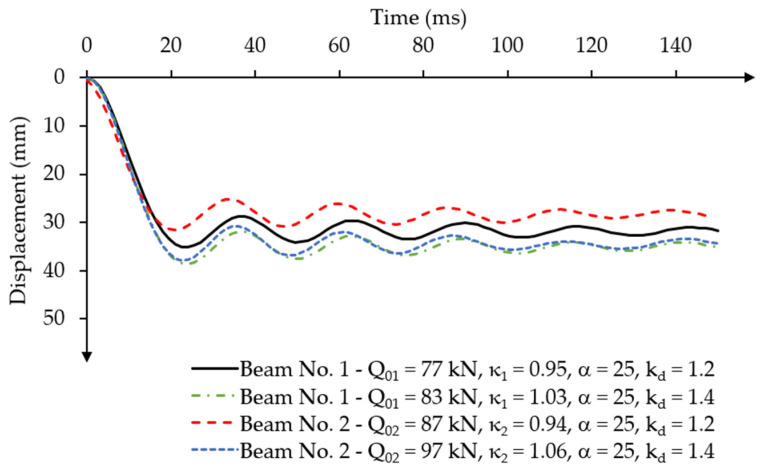
Results of the displacement-time relationship under constant force impulse loading over time, for different dynamic amplification factors for beam No. 1 and beam No. 2.

**Figure 15 materials-18-05432-f015:**
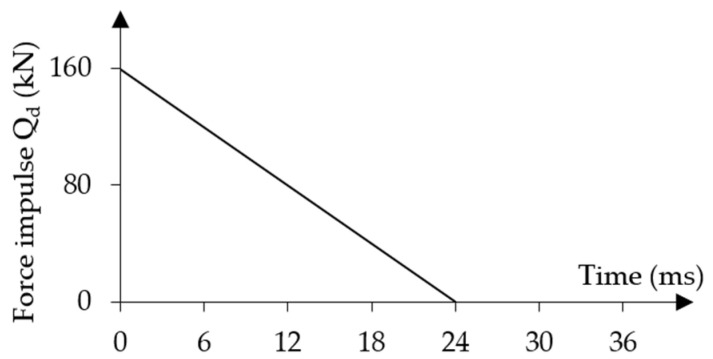
Characteristics of short-term time-varying force impulse loading.

**Figure 16 materials-18-05432-f016:**
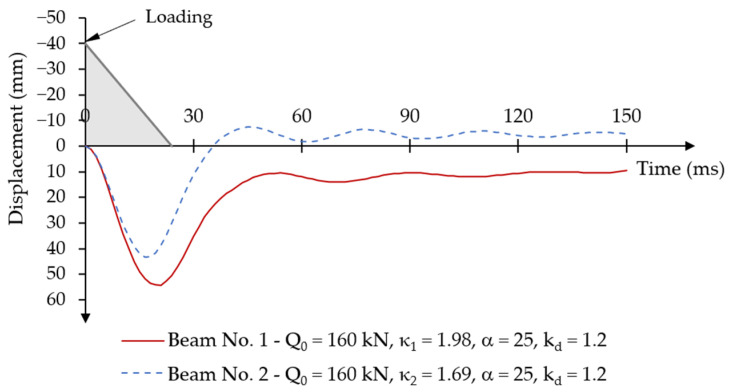
Results of the displacement-time relationship under short-term time-varying force impulse loading.

**Figure 17 materials-18-05432-f017:**
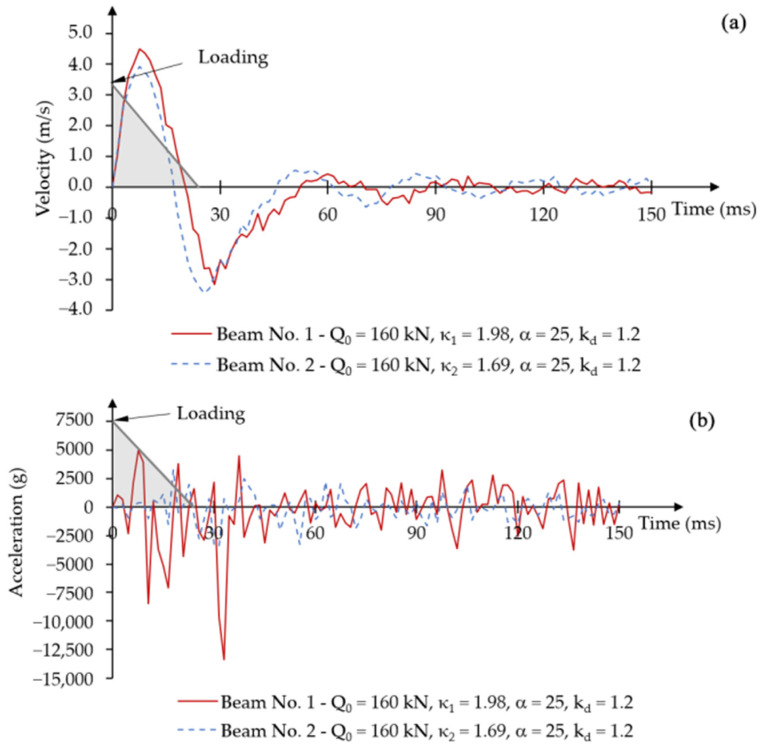
Loading with a short-term time-varying force impulse with damping: (**a**) velocity–time relationship; (**b**) acceleration–time relationship.

**Figure 18 materials-18-05432-f018:**
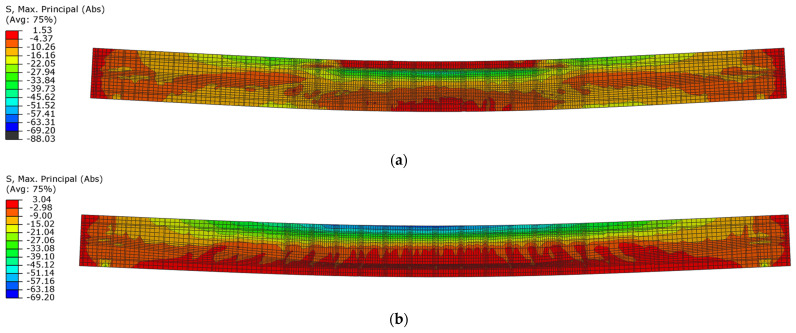
Maximum principal stresses in concrete in the first loading phase: (**a**) beam No. 1 at time t = 16.5 ms; (**b**) beam No. 2 at time t = 21 ms.

**Figure 19 materials-18-05432-f019:**
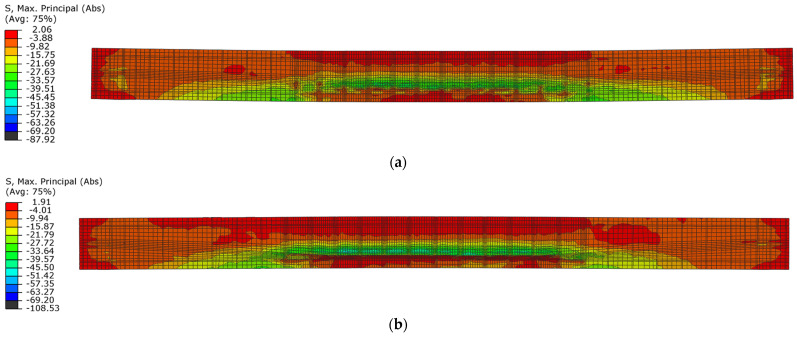
Maximum principal stresses in concrete in the first unloading phase: (**a**) beam No. 1 at time t = 54.0 ms; (**b**) beam No. 2 at time t = 45.0 ms.

**Figure 20 materials-18-05432-f020:**
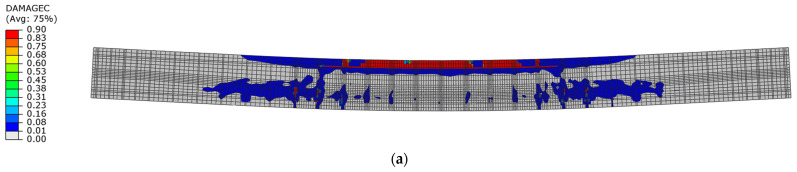
Compression damage in concrete in the first loading phase: (**a**) beam No. 1 at time t = 16.5 ms; (**b**) beam No. 2 at time t = 21 ms.

**Figure 21 materials-18-05432-f021:**
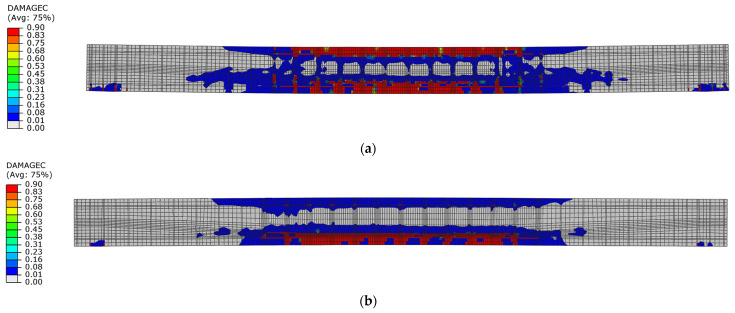
Compression damage in concrete in the first unloading phase: (**a**) beam No. 1 at time t = 54.0 ms; (**b**) beam No. 2 at time t = 45.0 ms.

**Figure 22 materials-18-05432-f022:**
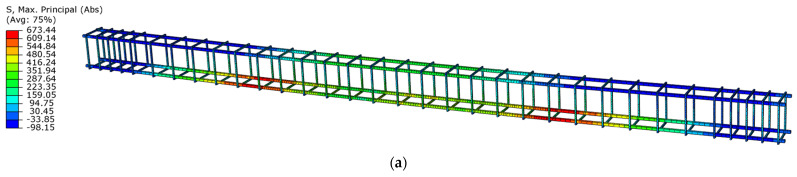
Maximum principal stresses in reinforcing steel in the first unloading phase: (**a**) beam No. 1 at time t = 54.0 ms; (**b**) beam No. 2 at time t = 45.0 ms.

**Figure 23 materials-18-05432-f023:**
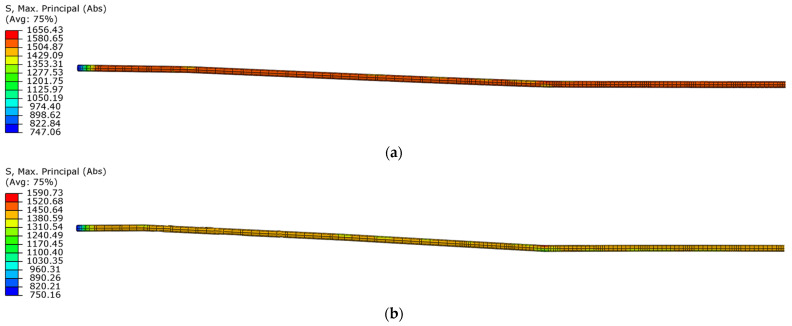
Maximum principal stresses in prestressing steel in the first unloading phase: (**a**) beam No. 1 at time t = 54.0 ms; (**b**) beam No. 2 at time t = 45.0 ms, (the drawing refers to the left half of the tendons to the beams’ axis of symmetry).

**Figure 24 materials-18-05432-f024:**
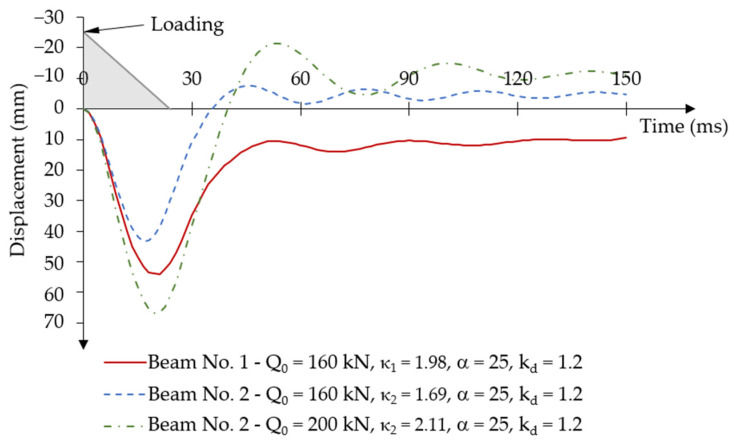
Results of the displacement-time relationship under various values of the short-term time-varying force impulse loading in beam No. 2.

**Figure 25 materials-18-05432-f025:**
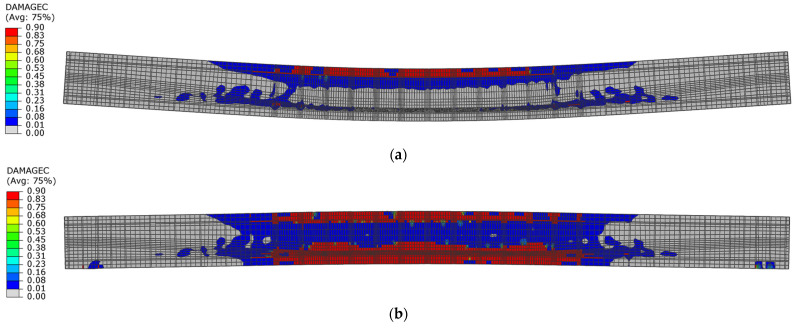
Compression damage in concrete of beam No. 2: (**a**) in the first loading phase at time t = 19.5 ms; (**b**) in the first unloading phase at time t = 54 ms.

**Figure 26 materials-18-05432-f026:**
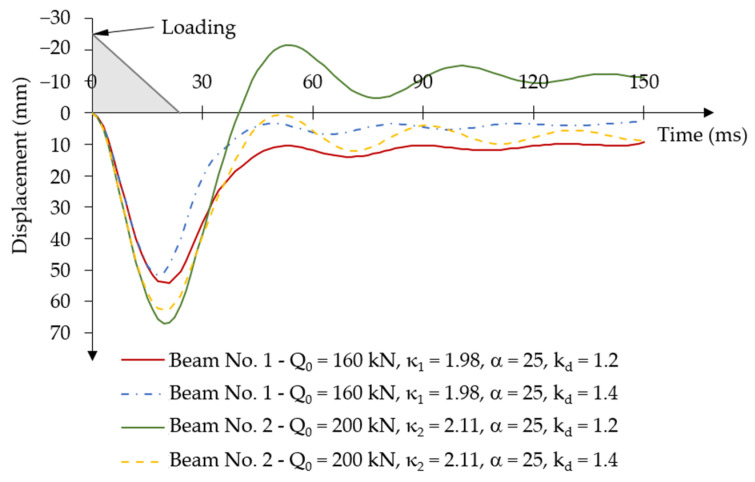
Results of the displacement-time relationship under short-term time-varying force impulse loading for various dynamic strength coefficients of concrete.

**Table 1 materials-18-05432-t001:** Concrete model parameters [9].

Parameter	Value		
	$k_{d}$ = 1.0	$k_{d}$ = 1.2	$k_{d}$ = 1.4
$f_{c}=f_{cm}$	59.0 MPa	69.20 MPa	79.40 MPa
$f_{ct}=f_{ctm}$	4.095 MPa	4.39 MPa	4.64 MPa
$E_{c}=E_{cm,eff}$	21.05 GPa	22.08 GPa	23.01 GPa
$\beta_{ce}=[0.2;\;0.5]$	0.4	0.4	0.4
$\beta_{cr}=[0.05;\;0.4]$	0.1	0.1	0.1
$\beta_{ctr}=[0.05;\;0.2]$	0.1	0.1	0.1
$\varepsilon_{cf}$	3.45‰	3.75‰	4.03‰
$\varepsilon_{ctf}$	0.19‰	0.20‰	0.20‰
$\varepsilon_{cr}=\left[6.0;\;12.0\right]$	6.0‰	6.60‰	7.20‰
$\varepsilon_{ctr}=\left[0.3;\;0.6\right]$	0.42‰	0.42‰	0.42‰

**Table 2 materials-18-05432-t002:** CDP model parameters.

Parameter	ψ (°)	ε	$\sigma_{b0}/\sigma_{c0}$	$K_{c}$	μ
Value	56.3	0.1	1.16	0.677	0

**Table 3 materials-18-05432-t003:** Material parameters of the Johnson–Cook model for steel elements.

Steel Element	E (GPa)	A (MPa)	B (MPa)	ρ (kg/m^3^)	ν	n	C	${\dot{\varepsilon }}_{0}$ $(s^{-1}$)	T (°K)	$T_{trans}$ (°K)	$T_{melt}$ (°K)	m
Prestressing steel	181.6	1816	7112	7850	0.3	0.024	0.02	0.00001	293	293	1540	1.03
Reinforcing bar ϕ6	210.0	532	1136				0.05					
Reinforcing bar ϕ10	210.0	541	943									
Anchorage	200.0	430	1710									

## Data Availability

The original contributions presented in this study are included in the article. Further inquiries can be directed to the corresponding authors.

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
