# Peer review of "Numerical Simulation of the Post-Tensioned Beams Behaviour Under Impulse Forces Loading"

_materials, 2025, doi:10.3390/ma18235432_

Round 1

Reviewer 1 Report

Comments and Suggestions for Authors

General Comment

The submitted manuscript investigates the dynamic behavior of post-tensioned concrete beams subjected to two types of impulse loading (constant force impulse and short-term time-varying impulse), utilizing detailed finite element numerical simulations performed in Abaqus/Explicit. The numerical models include concrete modeled with the Concrete Damage Plasticity (CDP) model and steel components modeled using the Johnson–Cook constitutive law, with the dynamic behavior of materials adjusted through dynamic strength coefficients. Prestressing tendon layout, geometry, material parameters, and damping characteristics are incorporated based on previously calibrated static experiments.

The study addresses a relevant research topic, as dynamic or impulsive actions on prestressed concrete elements are of increasing interest in structural engineering, particularly in the context of impact, blast, and accidental loading scenarios. The manuscript clearly presents the modeling approach, material characterization methodology, dynamic load application strategy, and resulting displacement, stress, and damage responses. The results show how loading type and prestressing eccentricity affect dynamic load-capacity and damage progression.

I made some suggestions and comments to improve the manuscript. The authors should take the suggestions into account, revise their manuscript and resubmit it.

Specific Comment 1

The manuscript requires careful and thorough English editing to improve readability, academic style, and clarity. There are numerous grammatical mistakes, unclear sentences, formatting inconsistencies, and awkward phrasing. A professional language editing service is strongly recommended.

Some examples:

- Capitalization of technical terms is inconsistent.

- Some paragraphs are overly long and difficult to follow.

- Figures vary in style and resolution.

- Equation numbering, symbol formatting, and variable definitions are sometimes unclear.

- …

Specific Comment 2

The abstract should be tightened to more clearly state:

- the research gap,

- the main methodological approach,

- the key quantitative findings,

- and the significance/novelty of the work.

Currently, it reads primarily as a description rather than a concise scientific summary.

Specific Comment 3

While the study is well executed, the novelty is not sufficiently highlighted. The introduction should include a clearer statement explaining:

- What specific gap in the literature this work addresses,

- How this study advances understanding beyond previous numerical or experimental research.

Consider adding a final paragraph in the introduction explicitly outlining novelty.

Specific Comment 4

The current literature review focuses on studies involving impact loading and prestressed structures, but misses several recent contributions on numerical modeling, CDP calibration strategies, strain-rate sensitive behavior of concrete, and dynamic response of PT beams.

The manuscript would benefit from adding and discussing recent sources (2022-2025)

These should be incorporated into the discussion of Section 1.

Specific Comment 5

Although the manuscript states that the models were calibrated based on static experiments, the validation discussion is limited. Please:

- Include direct comparative plots (load–displacement curves, crack patterns) between numerical and experimental outcomes.

- Clarify margin of deviation and model reliabiility.

Specific Comment 6

The selection of values kd=1.2k_d = 1.2kd=1.2 and kd=1.4k_d = 1.4kd=1.4 needs stronger justification. Please, provide clear references that relate these values to expected strain rate ranges for the loading conditions analyzed.

Specific Comment 7

The selection of Rayleigh damping parameter α = 25 is explained, but further justification is needed. Please, include:

- comparison of numerical free-vibration decay vs. experimental or analytical estimate,

- explanation of how sensitive the dynamic results are to this choice.

Specific Comment 8

Some figures require improvement, namely increase resolution for better readability (namely, figures with colour maps).

Specific Comment 9

The results section is descriptive. The discussion would benefit from:

- clear physical interpretation of mechanisms influencing differences between Beam 1 and Beam 2,

- linking observed failure patterns to prestressing eccentricity,

- addressing how numerical predictions may influence design or assessment practices.

Specific Comment 10

The conclusions are lengthy and read as result summaries. They should be reorganized into:

- main technical findings,

- practical implications for design/assessment,

- limitations,

- possible future research.

Comments on the Quality of English Language

See Specific Comment 1

Author Response

Best Regards,

Anna Jancy

Reviewer 2 Report

Comments and Suggestions for Authors

This paper presents a numerical simulation of the dynamic behavior of post-tensioned beams under constant time-invariant and short-term time-varying force impulse loads, using the Abaqus/Explicit procedure with calibrated models based on static experimental data. The results indicate that the dynamic load-capacity of the beams is approximately 5% lower than the static load-capacity under constant impulse loads, while it significantly exceeds the static load-capacity (up to 211% for the beam with larger prestressing eccentricity) under short-term time-varying impulse loads, with concrete dynamic strength coefficient and prestressing eccentricity exerting notable influences. This paper needs minor revision, the detailed comments are:

  1. The abstract effectively summarizes the core findings but could further emphasize the research gap addressed.
  2. The introduction provides a clear literature review but may briefly highlight how the proposed explicit procedure with Rayleigh damping parameter selection advances existing numerical modeling approaches.
  3. For the Johnson-Cook model used for steel elements, the dynamic hardening parameter C was determined approximately due to the absence of experimental strain rate data for the specific steel types. It is recommended to supplement sensitivity analysis of this parameter or cite more relevant studies to justify the adopted values, improving the model's reliability.
  4. The duration of the short-term time-varying impulse load (24 ms) is assumed without explicit justification. Providing a rationale for this selection (e.g., based on typical blast load durations or structural dynamic response characteristics) and conducting a parametric study on load duration would make the loading scheme more rigorous.
  5. The conclusions effectively summarize the key results but could be expanded to include practical engineering implications.

Author Response

Best Regards,

Anna Jancy

Reviewer 3 Report

Comments and Suggestions for Authors

Dear authors, the paper “Numerical simulation of the behavior of post-tensioned beams under the influence of impulse forces” presents a numerical analysis of the behavior of post-tensioned concrete beams subjected to two types of impulse loading (constant and short-term time-varying). This is an area with a limited number of studies available in the literature. You have identified a research gap - the lack of detailed models using explicit solution procedures and damping - and propose a numerical approach based on the Abaqus/Explicit method.

However, in its current form, the paper is not ready for publication and requires technical corrections and methodological improvements. In this context, I make the following comments and suggestions:

In the introductory part, the role of prestressing and the effects and consequences of dynamic loading are explained at the textbook level, which is not necessary in papers intended for publication in a special issue of a scientific journal.

The literature review is very concise, omitting studies that provide comprehensive 3D models for the nonlinear analysis of prestressed concrete structures. Moreover, for the presented references the advantages and limitations of their findings and/or results are not discussed. The state of the art should present a clear picture of the current research in order to highlight the scientific contribution and originality of the proposed work.

Furthermore, the work does not show an original scientific contribution; rather, it represents a synthesis and combination of existing models. Although this is acceptable, originality should be reflected in the interpretation of the results, which should also be checked. In this study, all results and conclusions are based solely on the numerical analysis of two beams. The model combines an embedded Abaqus model with Johnson-Cook parameters for steel members, but these parameters are derived only from static analyzes and lack experimental validation by dynamic testing, which limits the reliability of the model.

The presented conclusions are derived from a numerical model, with correlations with static analyses. It is recommended to validate the numerical results against dynamic test data - which can be taken from the existing literature - and to clearly state the limitations of the proposed model, since all conclusions should be treated with caution.

In addition to the aforementioned methodological improvements, technical corrections should be made, namely the correct numbering of equations, since equation (6) is followed again by equation (5), then equations (6), (7), etc. Accordingly, all references to equations in the text should be revised.

I hope that my suggestions will help improve your work.

Best regards

Author Response

Best Regards,

Anna Jancy

Reviewer 4 Report

Comments and Suggestions for Authors

Please find an attached.

Author Response

Best Regards,

Anna Jancy

Round 2

Reviewer 1 Report

Comments and Suggestions for Authors

I received the revised version of the review article with title “Numerical Simulation of the Post-Tensioned Beams Behaviour under Impulse Forces Loading” and also the authors’ responses and clarifications to my previous comments. The authors have improved the article according to most of my previous comments and suggestions, and I’m also satisfied with the given clarifications. I’m globally satisfied with the new version of the article. I recommend that the article should be accepted for publication.

Reviewer 3 Report

Comments and Suggestions for Authors Dear authors, You have mostly accepted the recommendations which I hope have improved your article.
Best regards

Reviewer 4 Report

Comments and Suggestions for Authors

This reviewer's comments were well reflected.